# Comparative Safety Profiles and Usage Patterns of Iodinated Contrast Media in Medical Imaging

**DOI:** 10.3390/diagnostics14222487

**Published:** 2024-11-07

**Authors:** Yu Ri Shin, Seo Yeon Youn, Hokun Kim, Ho Jong Chun, Hwa Young Lee, Hyo Joon Kim, Soon Nam Oh

**Affiliations:** 1Department of Radiology, Seoul St. Mary’s Hospital, College of Medicine, The Catholic University of Korea, Seoul 07345, Republic of Korea; crystal57@catholic.ac.kr (Y.R.S.); imseoyeon@gmail.com (S.Y.Y.); walehn@gmail.com (H.K.); hojongchun@gmail.com (H.J.C.); 2Department of Internal Medicine, Division of Allergy, Seoul St Mary’s Hospital, College of Medicine, The Catholic University of Korea, Seoul 06591, Republic of Korea; lehwyo@catholic.ac.kr; 3Department of Emergency Medicine, Seoul St. Mary Hospital, College of Medicine, The Catholic University of Korea, Seoul 06591, Republic of Korea; khjoon0110@gmail.com

**Keywords:** contrast media, incidence, premedication, tomography, X-ray computed

## Abstract

Objectives: This study aimed to analyze the usage patterns and hypersensitivity reaction (HSR) profiles of six nonionic iodinated contrast media (ICMs) used in computed tomography (CT) to enhance patient safety and inform evidence-based contrast agent selection. Methods: We retrospectively reviewed 248,209 CT scans obtained between January 2020 and December 2022. Six ICMs (iomeprol, iohexol, ioversol, iopromide, iodixanol, and iobitridol) were compared on the basis of their usage rates, HSR incidence, and severity. This study also evaluated the impact of premedication protocol reinforcement and assessed the quarterly HSR rates. Results: Among the 248,209 CT scans, 1603 (0.65%) were associated with HSRs. Most HSRs were mild (86.2%), with moderate (10.9%) and severe (2.9%) reactions being less common. Four ICMs were used as first-line agents and two ICMs were used as second-line agents. The second-line agents, iobitridol and iodixanol, exhibited 7–8 times higher HSR rates compared to the first-line agents. A modified premedication protocol implemented in mid-2022 significantly reduced the incidence of moderate HSRs (*p* = 0.0075). The quarterly analysis indicated a trend in higher HSR rates in the first quarter and a statistically significant increase in severe HSRs in the third quarter (*p* = 0.033). Conclusions: These findings highlight the importance of tailored premedication protocols and a 7–8 times higher rate of HSR with second-line agents in contrast-enhanced imaging. Future research should focus on elucidating the mechanisms underlying these variations to further refine contrast agent selection and management strategies.

## 1. Introduction

Iodinated contrast media (ICMs) play a pivotal role in enhancing the diagnostic accuracy of computed tomography (CT) by improving the visibility of anatomical structures and pathological conditions. Despite its widespread use and benefits, ICM administration is associated with a risk of hypersensitivity reaction (HSR), which can range from mild symptoms such as urticaria to severe and potentially life-threatening anaphylaxis. The advent of nonionic, low-osmolar, and iso-osmolar ICMs has significantly reduced the incidence of HSRs compared to earlier high-osmolar agents [1]. However, as the use of contrast-enhanced CT continues to expand globally, the absolute number of ICM-related HSRs remains a critical clinical concern [2,3].

Recent studies have emphasized the variability in safety profiles among different ICMs, suggesting that factors beyond osmolarity and ionicity, such as molecular structure and side chains, may influence the antigenicity of these agents [4]. For example, research has indicated that specific structural components of the ICMs could contribute to the likelihood of HSRs; however, these mechanisms are not yet fully understood. Additionally, patient-related factors, including a history of HSRs to ICMs, have been recognized as significant predictors of future reactions [5]. However, the relative contribution of these factors and their interactions remain unclear. Moreover, the existing literature on the comparative safety profiles of various ICMs is limited and often lacks direct head-to-head comparisons in large and diverse patient populations [6,7]. The potential influence of external factors, such as seasonal variations, on the incidence of HSRs remains controversial, with conflicting evidence reported in previous studies [8,9]. These gaps in knowledge hinder the development of evidence-based guidelines for contrast agent selection and the management of patients at risk of HSRs [10]. Given the unpredictability of ICM-related HSRs, it is crucial to systematically prevent and manage these reactions by accurately identifying their incidence and associated factors [11].

To address these gaps, this study aimed to provide a comprehensive analysis of the incidence and severity of HSRs associated with six commonly used nonionic ICMs over a three-year period at a large tertiary care center. Over the past three years, our institution has undergone several alterations in the use of contrast agents, including the incorporation of a new ICM and the modified implementation of premedication protocols. The objective of this study was to assess the influence of these modifications on the prevalence of adverse reactions to contrast agents. In addition, other potential factors affecting HSR incidence and severity were investigated. By utilizing a large dataset with standardized adverse event reporting, our goal was to provide evidence-based information that can assist in clinical decision-making regarding ICM selection and contribute to the development of safer imaging practices.

## 2. Materials and Methods

### 2.1. Patient Population

This retrospective cohort study was conducted at a large tertiary care institution, and the electronic medical records of patients who underwent contrast-enhanced CT between January 2020 and December 2022 were analyzed. The study was approved by the Institutional Review Board, and the requirement for informed consent was waived due to the retrospective nature of the study. All patients who underwent contrast-enhanced CT during the study period were included. Exclusion criteria were established to minimize potential bias and include cases with incomplete medical records or ambiguous reports of adverse reactions.

### 2.2. CT Contrast Media and Injection Protocol

Six nonionic ICMs were compared: iomeprol, iohexol, iopromide, iobitridol, ioversol, and iodixanol. Iodixanol was the only iso-osmolar agent in the group, whereas the others had low osmolarity. An ICM was assigned based on standard institutional protocols, except in cases with a documented history of HSR where an alternative ICM was selected. From 2020 through the first quarter of 2021, five contrast agents were used: iomeprol, iohexol, iopromide, iodixanol, and ioversol, with iodixanol only used as an alternative second-line contrast agent in patients with a documented history of HSR. In the second quarter of 2021, iobitridol was newly introduced as an alternate contrast agent, with iodixanol remaining the first-line agent along with the four other contrast agents from this point forward. In this study, first-line agents referred to as ICMs routinely used for patients without documented HSRs were selected based on standard institutional protocols. Second-line agents were designated as first-line agents for patients with a known history of HSRs. The utilization of second-line agents is aimed at minimizing the risk of HSR recurrence by substituting ICMs exhibiting different antigenic profiles when appropriate. The dose of contrast medium was determined according to the patient’s body weight and the specific CT protocol and ranged from 80 to 120 mL. Administration was performed using an automatic injector at a flow rate of 1.5–5 mL/s, depending on the protocol.

### 2.3. Hypersensitivity Reaction Classification and Premedication Protocol

HSRs were identified and classified using a standardized adverse event reporting system integrated into electronic medical records. Radiologists and nurses were trained to promptly recognize and document HSRs. Each reported HSR was reviewed and confirmed by a radiologist using criteria based on the American College of Radiology Manual on Contrast Media and Korean Clinical Practice Guidelines [12,13]. The mild category included symptoms such as limited urticaria/pruritus, cutaneous edema, limited itchy/scratchy throat, nasal congestion, and sneezing/conjunctivitis/rhinorrhea. The moderate category included symptoms such as diffuse urticaria/pruritus, facial edema without dyspnea, throat tightness or hoarseness without dyspnea, and wheezing/bronchospasm with mild or no hypoxia. The severe category included symptoms such as diffuse edema or facial edema with dyspnea, diffuse erythema with hypotension, laryngeal edema with stridor and/or hypoxia, wheezing/bronchospasm with significant hypoxia, and anaphylactic shock.

The institutional premedication protocol was based on the severity of any previous HSR. An antihistamine was administered for mild reactions. For moderate reactions, a combination of an antihistamine and a single-dose steroid was used; for severe reactions, avoidance of ICM was recommended, if possible. A CT scan should be performed at least 30 min after premedication to optimize the efficacy of corticosteroids. However, in the process of checking our response system to contrast HSR in early 2022, we found that a CT scan was performed immediately after the injection of premedication. Therefore, we thoroughly trained nurses to perform CT scans at least 30 min after premedication injection and compared the HSR rates before and after this time.

### 2.4. Data Collection and Statistical Analyses

Data were extracted from electronic medical records, including patient demographics, ICM type and dose, history of previous ICM exposure and reaction, premedication use, occurrence and severity of HSRs, and HSR rate according to the quarter.

Descriptive statistics were used to summarize the data. The incidence of HSRs was calculated as a proportion of the total number of CT scans. Comparisons of HSR rates between different contrast media were conducted using the Kruskal–Wallis test for non-parametric data and ANOVA for parametric data. The Wilcoxon rank-sum test was used to compare the distribution of HSR severity between specific ICM groups, particularly when the first- and second-line agents were compared. To assess the impact of the premedication protocol with a strict delay in the CT scan, we performed a paired *t*-test to compare HSR rates before and after strict protocol adherence. The Cochran–Armitage trend test was used to analyze quarterly variations in HSR rates. Subgroup analyses were conducted to explore the differences in HSR rates between the first- and second-line contrast agents. To account for the potential confounding effect of iodixanol use as a second-line agent, we performed separate analyses for the periods before and after the first quarter of 2021, when its use pattern changed. Statistical significance was defined as a *p*-value of less than 0.05. All analyses were performed using SAS version 9.4 (SAS Institute, Cary, NC, USA) and R software, version 4.2.2.

## 3. Results

### 3.1. HSR Incidence and Severity by Contrast Medium

In total, 248,209 contrast-enhanced CT scans were obtained during the study period. The mean age of the patients was 63 ± 15.45 years (median, 65 years), with a balanced sex distribution (49.9% male). Table 1 summarizes the usage frequency and HSR incidence of each ICM. The overall incidence of HSR was 0.65% (1603 cases). HSR rates differed significantly among ICMs (*p* < 0.001): 3.6% for iobitridol, 2% for iodixanol, 0.87% for iomeprol, 0.82% for iopromide, 0.42% for iohexol, and 0.4% for ioversol (Table 1; Figure 1).

The data were separated from the first quarter of 2021 onwards because iodixanol was used as the second-line agent until the first quarter of 2021, after which iobitridol and iodixanol were used as the second- and first-line agents, respectively. The HSR rates for iodixanol and iobitridol, which were used as alternative agents in both periods, were 4.4% and 3.8%, respectively (Figure 2A,B). These rates were significantly higher than the mean HSR rates of 0.59% and 0.72% for first-line agents in each time period (*p* = 0.0085 and *p* < 0.0001, respectively). A comparison of the two second-line agents revealed that iobitridol exhibited a slightly lower HSR rate than iodixanol (3.8% vs. 4.4%, respectively). However, this difference was not statistically significant (*p* = 0.5714).

Among the patients who experienced HSR, 86.2% (*n* = 1382), 10.9% (*n* = 175), and 2.9% (*n* = 46) had mild, moderate, and severe diseases, respectively. When comparing HSR severity by ICM, mild and moderate HSR differed between contrast agents (*p* < 0.0001), but severe HSR did not differ between contrast agents (*p* = 0.2614) (Table 2, Figure 3).

### 3.2. Effects of Premedication Protocol Reinforcement: Strict Training for Delay of CT Scan After Premedication

To optimize the efficacy of premedication, CT scanning should be properly delayed about one hour after premedication injection. However, upon realization of the insufficient temporal delay due to tight CT scheduling, the training of nurses commenced in the second quarter of 2022 to ensure that patients with a history of HSR received a delayed CT scan for a minimum of 30 min after premedication. Following the modification of the rigorous premedication protocol, the overall HSR rate decreased from 0.68% to 0.55% (*p* = 0.0979). Although this overall change was not statistically significant, there was a significant decrease in moderate HSRs (0.08% to 0.03%, *p* = 0.0075), whereas there was little change in severe HSRs (Table 3). All ICM agents also showed a reduction in HSR rate, but only iobrix showed a statistically significant reduction (*p* = 0.0409).

### 3.3. Quarterly Trends in HSR Incidence

We observed a trend towards higher HSR rates in the first quarter of each year, decreasing towards the fourth quarter, although this did not reach statistical significance (*p* = 0.9471) (Table 4, Figure 4A). When comparing HSR severity by quarter, we found a statistically significant increase in severe HSR in the third quarter (*p* = 0.033) (Table 5; Figure 4A).

When quarterly variation was analyzed separately for first- and second-line agents, the HSR rates for second-line agents showed a decreasing trend from the first to the fourth quarter, which was more pronounced than that for first-line agents, although this was not statistically significant (Table 6, Figure 4B). While not an exact match, Korea has four relatively distinct seasons, with the first/second quarter corresponding to winder/late spring and the third/fourth quarter corresponding to summer/late fall.

## 4. Discussion

This retrospective study provides comprehensive data on the safety profiles and usage patterns of six nonionic ICMs in a tertiary care setting. Our findings revealed substantial differences in HSR rates among these agents, with important implications for clinical practice and patient safety. The overall incidence of HSR in our study (0.65%) was relatively low compared with that in earlier reports, consistent with the trend in decreasing adverse events following the transition to nonionic, low-osmolar ICMs [7,14]. The HSRs identified in this study were 0.4–0.8%, which is a relatively low incidence compared with other studies. Differences in study timing may explain the lower incidence of HSRs in this study compared to the previous studies. This may be due to the fact that the adverse event reporting system has become more reliable than in the past, and patients who have experienced HSR have been able to substitute other contrast agents and actively use pretreatment. In previous studies, not only allergic reactions, but also physiological reactions, such as vomiting and headaches, were comprehensively included as side effects of contrast agents. In our study, the incidence of side effects was mainly allergic reactions, which seems to be due to the difference in the range recognized as a symptom of HSR [6,15]. Similar to previous studies, our study showed that iobitridol and iodixanol, used as second-line contrast agents, had a 7–8 times higher rate of HSR than other first-line contrast agents, suggesting that multiple exposures to contrast agents further increase the likelihood of HSRs [5,16,17,18].

Although the difference between iodixanol and iobitridol was not statistically significant, the HSR rate showed a slight decrease from 4.4% to 3.81% when iobitridol was used as a second-line agent, suggesting a potential trend towards improved safety. This finding aligns with current research highlighting the role of chemical side chains in HSR associated with ICMs [4]. Structural differences between contrast agents are significant factors that influence their safety profiles. Specifically, the side chains of molecules play a critical role in their antigenicity and potential to cause HSR. Iodixanol contains an N-(2,3-dihydroxypropyl)-carbamoyl side chain, which has been identified as an antigenic determinant contributing to cross-reactivity among ICMs. This side chain is shared by a group of ICMs, increasing the likelihood of HSRs in patients with a history of reaction to these agents. Conversely, iobitridol has a different side chain, the N-(2,3-dihydroxypropyl)-N-methyl-carbamoyl moiety, which is less common than that in other ICMs. The switch to iobitridol likely reduced the recurrence of HSRs due to the absence of a common antigenic side chain in iodixanol and other similar agents. This substitution is supported by the findings of the study, which showed a significant reduction in HSR recurrence rates when using an ICM without a common side chain compared to those with the same or similar side chains [16].

Research on patient-related risk factors has revealed that prior ICM-related HSRs are the most significant predictors of future ICM-related HSRs [5,19]. Three main strategies have been widely applied to manage patients with previous HSRs according to severity: premedication, changes in culprit ICMs, and skin tests. In our study, until the first quarter of 2022, most premedications were administered immediately before re-exposure to ICMs. However, considering the pharmacological action time of corticosteroids, it is possible that the premedication in our study may not have had enough time to take effect [20]. Therefore, we trained the nurses to adhere to a sufficiently delayed CT scan after pretreatment completion. The change in our institution’s premedication protocol in the second quarter of 2022 is associated with a reduction in the overall HSR rates, particularly for moderate reactions. Although not statistically significant for all severity levels, there was a notable reduction in HSR incidents, which can be attributed to changes in the pretreatment method. While premedication has long been a cornerstone of HSR prevention, our findings suggest that adherence to standardized evidence-based pretreatment protocols, including a sufficiently delayed CT scan, may further reduce HSR incidence.

Although our findings are based on quarterly data rather than a strict reflection of the four seasons, they still reveal intriguing seasonal patterns in HSR rates to ICMs, with a higher incidence in the first quarter and a gradual decline towards the fourth quarter. This consistent trend suggests potential underlying factors influencing HSR incidence throughout the year, aligning with observations in other allergic and immunological conditions such as allergic rhinitis and asthma exacerbations, which often peak in spring and early summer. While these quarterly differences did not reach statistical significance, a notable finding was the statistically significant increase in severe HSR during the third quarter. This suggests that, while their impact on the overall incidence is less clear, seasonal factors may influence the severity of HSR. There has been an ongoing debate regarding seasonal variations in contrast medium reactions. Previous studies by Mortelé et al. and Callahan et al. found no significant correlation between HSR and seasonal time, whereas Ho et al. and Mikkonen et al. observed seasonal variations in HSR [8,9,21,22]. Our study suggests a potential seasonal pattern in HSR with different peak seasons compared with previous reports. However, the mechanisms underlying this seasonal pattern remain unclear. The potential factors include environmental allergens, medication patterns, and mast cell reactivity. Our findings have several clinical implications. Awareness of the increased HSR risk during certain seasons could inform tailored premedication strategies, enhanced monitoring protocols, or the consideration of alternative imaging modalities during high-risk periods. The observed seasonal patterns emphasize the need for year-round vigilance in ICM administration and HSR management. Further studies are required to explore the mechanisms underlying seasonal variations. Prospective studies incorporating environmental data, patient-specific factors, and immunological profiles may provide season-specific risk assessment tools and prevention strategies to further enhance patient safety during contrast-enhanced imaging.

Our study has several limitations that should be considered when interpreting the results. First, its single-center retrospective design may limit the generalizability of our findings to other health care settings. Multicenter prospective studies are valuable in confirming and extending our observations. Second, our inability to fully account for patients’ prior contrast exposure at other facilities introduces potential confounding factors into our analysis of HSR risk. Nevertheless, whenever a history of contrast exposure was identified, such as premedication before the first CT scan or adverse reactions to contrast agents reported by the patient, it was incorporated into the patient’s contrast exposure history to the greatest extent possible. Finally, although our adverse event reporting system was standardized, there is always the potential for the under-reporting or misclassification of mild reactions. The implementation of automated detection systems or prospective monitoring protocols could enhance the accuracy of HSR identification in future studies.

Despite these limitations, our findings have several important implications for clinical practice. First, the higher HSR rates associated with second-line agents in patients with previous reactions highlight the need for careful monitoring and the consideration of alternative imaging modalities for high-risk patients. Second, our data support the use of standardized premedication protocols with sufficient intervals before ICM administration, particularly to reduce moderate HSRs. We believe that this case illustrates the need not only to establish an appropriate system for responding to HSR but also to constantly monitor the system to ensure that it is well maintained.

In conclusion, this study provides valuable insights into ICM safety profiles for optimizing patient care. Our findings highlight the importance of agent-specific considerations, strict adherence to and monitoring of optimized premedication protocols, and awareness of quarterly variations in HSR risk. Future research should focus on elucidating the mechanisms underlying these observations, particularly the roles of chemical structures and environmental factors in ICM-related HSRs. By incorporating these insights into clinical practice, health care providers can enhance patient safety and maximize the diagnostic benefits of contrast-enhanced imaging.

## Figures and Tables

**Figure 1 diagnostics-14-02487-f001:**
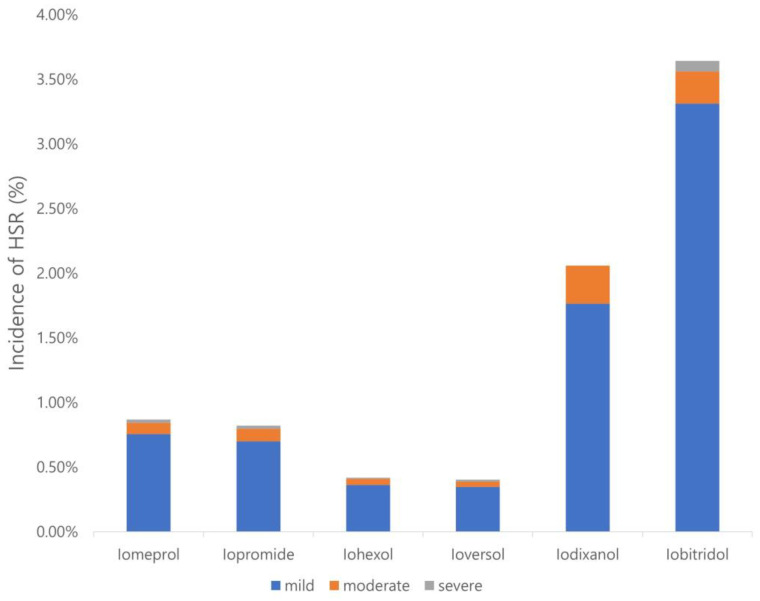
The overall incidence and severity of HSR according to the generic profile of ICMs.

**Figure 2 diagnostics-14-02487-f002:**
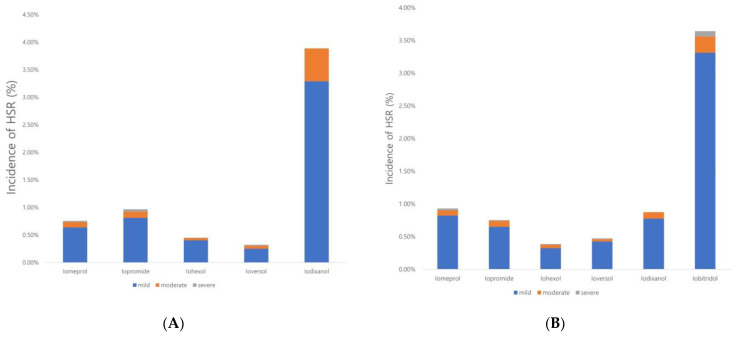
Incidence and severity of HSR before (**A**) and after 1st quarter of 2021 (**B**).

**Figure 3 diagnostics-14-02487-f003:**
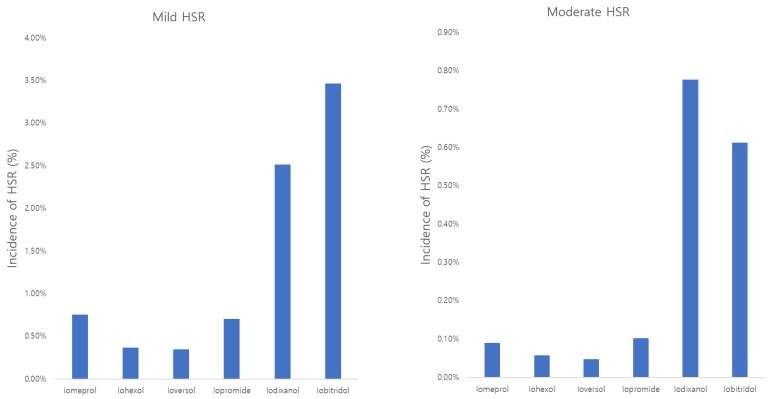
Difference in HSR severity according to contrast agent.

**Figure 4 diagnostics-14-02487-f004:**
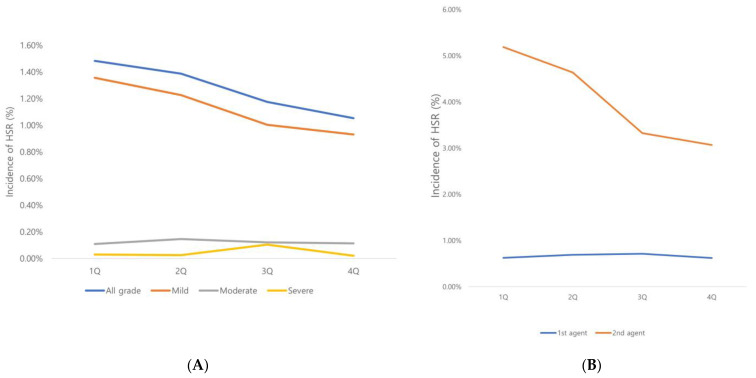
(**A**). Difference in HSR severity according to contrast agent. (**B**) Quarterly variation in the HSR rate according to first- and second-line contrast agents.

**Table 1 diagnostics-14-02487-t001:** Incidence and severity profile according to the individual agents.

Contrast Agent	Total Usage	HSR Incidence	HSR Rate (%)	*p*-Value	Mild HSR	Moderate HSR	Severe HSR
Iomeprol	56,665 (22.8)	492	0.87	<0.0001	428 (87)	49 (9.96)	15 (3.05)
Iohexol	62,358 (25.1)	261	0.42		225 (86.2)	30 (11.49)	6 (2.3)
Ioversol	63,523 (25.6)	256	0.4		220 (85.9)	26 (10.16)	10 (3.91)
Iopromide	62,755 (25.3)	515	0.82		439 (85.2)	61 (11.84)	15 (2.91)
Iodixanol	1700 (0.7)	35	2.06		30 (85.7)	5 (14.29)	
Iobitridol	1208 (0.5)	44	3.64		40 (90.9)	3 (6.82)	1 (2.27)

Values are presented as numbers, with percentages in parentheses.

**Table 2 diagnostics-14-02487-t002:** Severity of HSR for each contrast agent.

Severity	Contrast Agent	Total Usage	HSR Incidence	HSR Rate (%)	*p*-Value
Mild	Iomeprol	56,665	428	0.76	
	Iohexol	62,358	225	0.37	
	Ioversol	63,523	220	0.35	<0.0001
	Iopromide	62,755	439	0.7	
	Iodixanol	1700	60	2.51	
	Iobitridol	1208	40	3.46	
Moderate	Iomeprol	56,665	49	0.09	
	Iohexol	62,358	30	0.05	
	Ioversol	63,523	26	0.04	<0.0001
	Iopromide	62,755	61	0.1	
	Iodixanol	1700	5	0.78	
	Iobitridol	1208	3	0.61	
Severe	Iomeprol	56,665	15	0.04	
	Iohexol	62,358	6	0.02	
	Ioversol	63,523	10	0.02	0.2614
	Iopromide	62,755	15	0.03	
	Iodixanol	1700			
	Iobitridol	1208	1	0.6	

Values are presented as numbers.

**Table 3 diagnostics-14-02487-t003:** Effects of premedication protocol reinforcement: strict training for delay of CT scan.

	HSR Rate Before Training (Total Usage/HSR Incidence)	HSR Rate After Training (Total Usage/HSR Incidence)	*p*-Value
All grade	0.68% (177,945/1227)	0.55% (70,264/376)	0.0979
Mild	0.58% (177,945/1037)	0.51% (70,264/345)	0.2617
Moderate	0.08% (177,945/153)	0.03% (70,264/21)	0.0075
Severe	0.02% (177,945/37)	0.02% (177,945/37)	0.2771
Iomeprol	0.89% (40,167/360)	0.83% (16,498/132)	0.6826
Iohexol	0.46% (43,106/202)	0.31% (19,252/59)	0.046
Ioversol	0.42% (43,161/184)	0.36% (20,362/72)	0.3795
Iopromide	0.84% (49,611/419)	0.76% (13,144/96)	0.4955
Iodixanol	3.01% (1209/33)	0.4% (491/2)	0.0664
Iobitridol	4.16% (691/29)	3.35% (517/15)	0.6149

Values are presented as numbers.

**Table 4 diagnostics-14-02487-t004:** Quarterly variation in HSR incidence.

Quarter	Total Usage	HSR Incidence	** HSR Rate (%)	*p*-Value
1	57,473	368	1.48	0.9471
2	60,129	387	1.39	
3	59,752	415	1.18	
4	70,855	433	1.05	

Values are presented as numbers. ** HSR rates are calculated by the average of the HSR rate of each quarter.

**Table 5 diagnostics-14-02487-t005:** Quarterly variation by HSR severity.

Severity	Quarter	Total Usage	HSR Incidence	** HSR Rate (%)	*p*-Value
Mild	1	57,473	314	1.36	0.9744
	2	60,129	332	1.23	
	3	59,752	354	1.0	
	4	70,855	382	0.93	
Moderate	1	57,473	41	0.11	0.9979
	2	60,129	43	0.15	
	3	59,752	47	0.12	
	4	70,855	43	0.11	
Severe	1	57,473	13	0.03	0.033
	2	60,129	12	0.02	
	3	59,752	14	0.11	
	4	70,855	8	0.02	

Values are presented as numbers. ** HSR rates are calculated by the average of the HSR rate of each quarter.

**Table 6 diagnostics-14-02487-t006:** Comparison the quarterly variation of the HSR rate according to first- and second-line contrast agents.

Contrast Agent	Quarter	Total Usage	HSR Incidence	** HSR Rate (%)	*p*-Value
First-line agent	1	57,055	350	0.63	0.8632
	2	59,763	370	0.69	
	3	59,296	399	0.71	
	4	70,218	414	0.62	
Second-line agent	1	418	18	5.19	0.3706
	2	366	17	4.64	
	3	456	16	3.33	
	4	637	19	3.07	

Values are presented as numbers. ** HSR rates are calculated by the average of the HSR rate of each quarter.

## Data Availability

The datasets generated and/or analyzed during the current study are available from the corresponding author upon reasonable request.

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
