# Peer review of "Comparative Safety Profiles and Usage Patterns of Iodinated Contrast Media in Medical Imaging"

_diagnostics, 2024, doi:10.3390/diagnostics14222487_

Round 1

Reviewer 1 Report

Comments and Suggestions for Authors

The work of Y. R. Shin et al. brings an overview report on the analysis of patients’ reactions to the application of various iodinated contrast agents in computed tomography examination. The authors cover the utilization of the agents over several years and statistically evaluated nearly a quarter of a million cases. Such an evaluation is very relevant as it can bring useful recommendations for the future, therefore, it deserves publication. However, before the publication, the authors should check some presented results, as there seem to be some numerical mistakes. Also, some points should be in my opinion clarified.

I recommend to explicitly define what “first-line” and “second-line” agents are. When was used the second-line agent? Only when some HSR was observed in the first examination? Was premedication applied only when HSR was observed in the first application? Was sometimes the CT examination repeated with the same contrast agent, even if HSR was observed after the first application, and pre-medication was applied for the repeated examination? It is not so clear.

The main objection is related to the conclusion about seasonal dependence of HSR occurrence. The severe reaction was observed in 14 cases from 59752, which is not 0.11 %, but is similar to other quartiles (e.g. 13 from 57473 in the 1st, 12 of 60129 in 2nd). Obviously, some numerical mistake occurred (wrong number of examinations or occurrence of HSR, or calculation). Please, check all the data/calculations, and change the discussion accordingly.

Reviewer 2 Report

Comments and Suggestions for Authors

1. The reviewed article presents the results of a study comparing six nonionic iodinated contrast substances used in CT, analyzing their patterns of usage and also their association with hypersensitivity reactions. 

2. The article is structured correctly into the classic parts of an original article.

Although the subject is a common one, this is a vast retrospective study bringing valuable data to the current medical context, which is accurately described in the introduction.

Material and methods are very clearly described and can be understood by anyone, no matter of medical specialty. The authors reviewed a significant number of CT scans (248209- performed between 01.2020 and 12.2022 in a tertiary care center) and compared the six contrast media used in these CT regarding rates of usage and the incidence of hypersensitivity reactions, their severity and the impact on premedication on hypersensitivity reactions. Please provide some supplementary data regarding the indications of the CTs and maybe if there was an  initial  indication for a specific contrast media (eg. Iohexol was preffered for pulmonary contrast...)

Results are well formulated. The authors identified four contrast media as first line agents and two as second line agents and established an association between usage and hypersensitivity reactions, revealing the fact that second line agents are associated with a significantly hypersensitivity risk compared to first line agents. Importantly, premedication protocols significantly reduced the incidence of hypersensitivity, reassuring clinical and imaging doctors that their tools for performing safe contrast CT s are working properly. Tables and Figures reflect the results well, I know this is a retrospective study focused on imaging but I would like to have a short description of the patients having HSR and why they had their CTs. 

The study has two main conclusions, very useful for clinical practice: 1. contrast agents are divided into first and second line according to usage, which is conditioned by hypersensitivity reactions; 2. Premedication diminishes in clinical practice the incidence of hypersensitivity and contrast CT can now be performed safely in patients prone to allergies.

Discussion are well conducted, clearly written and documented. The authors propose also some directions regarding future research.

7. The references are adequate, the majority being from the last 5 years and many from the last 2 years.

8. I didn’t detect any significant grammar or spelling errors, English language is fluid.
